# *Opuntia* spp. as Alternative Fodder for Sustainable Livestock Production

**DOI:** 10.3390/ani12131597

**Published:** 2022-06-21

**Authors:** Grazia Pastorelli, Valentina Serra, Camilla Vannuccini, Everaldo Attard

**Affiliations:** 1Department of Veterinary Medicine and Animal Sciences, University of Milano, Via dell’Università 6, 26900 Lodi, Italy; cami.vannuccini@gmail.com; 2Division of Rural Sciences and Food Systems, Institute of Earth Systems, University of Malta, MSD 2080 Msida, Malta; everaldo.attard@um.edu.mt

**Keywords:** prickly pear, arid regions, nutrition, animal performance

## Abstract

**Simple Summary:**

Livestock production is a significant subsector of agriculture around the world and can be adversely affected by the scarcity of good quality green forage. Among the main constraints on green forage production are erratic rainfall, water scarcity, and natural disasters due to climate change. Due to its adaptability to extreme conditions, cactus is a valuable fodder alternative for arid and semiarid regions of the world, where water represents a limiting factor for animal production. This review focuses on the *Opuntia* genus, and an overview of several studies on its use in livestock is presented.

**Abstract:**

During the past decades, livestock production has increased significantly, which has led to the degradation of rangelands due to overgrazing. The lack of water in several arid areas has led to a decline in crop and animal husbandry. As a consequence, the demand for drought-resistant crops has increased significantly so as to keep crop and animal husbandry systems viable and sustainable. *Cactaceae* have adaptive characteristics that ensure their development progress under drought conditions. The present review provides information on the nutritive value of *Opuntia* in animal fodder production, its effects on animal performance, and the quality of the animal-derived products. In conclusion, the use of *Opuntia* as innovative alternative feed would render animal production systems more sustainable.

## 1. Introduction

Livestock production is a significant subsector of agriculture around the world; it contributes to about forty percent of the total agricultural activities worldwide. In industrialized countries, livestock product contributes to more than half of this. The rise in livestock production increases the demands on the nutrient requirements to feed animals.

Livestock-based food production is facing a number of challenges partly attributed to climatic pollution, particularly to emissions, and also related to the competition between food, fodder, and fuel crops for the available agricultural land.

The pattern of forage supply varies from year to year, and crop production is unreliable in regions characterized by water scarcity [1], which is aggravated by changes in the climate. Consequently, the soil water deficit is increasing as the agricultural water demand is increasing (Figure 1) [2]. With the rapid changes in climate, the solutions emerging to solve this dryland situation are expected to be applied more extensively elsewhere over time. Therefore, solutions to maximize productivity and improve ecosystem health are here to stay [3]. With the very limited land allocation for fodder cultivation, water scarcity, and frequent drought-like conditions in many parts of the world where there is a huge livestock population, the production of sufficient fodder is a challenge for the scientific and farming community everywhere.

A typical example is the West Asia/North Africa (WANA) region, which is composed of large areas with rainy winters and hot dry summers. WANA is characterized by its high population growth, low and erratic rainfall, limited areas of arable land, harsh deserts, and limited water resources for irrigation development [4]. In India, the government has designated 30 percent of the country’s surface as wasteland [5]. In fact, in hot dry zone of India, the rural poor and smallholders are seriously affected by droughts and desertification. The future of the arid and semiarid zones depends on the development of sustainable agricultural systems and on the cultivation of appropriate crops able to survive calamities, such as drought, extreme temperatures, and poor soils. Especially in the described context, there is considerable interest in the *Cactaceae* family, within which the genera *Opuntia* and *Nopalea* were domesticated a long time ago and have been used ever since as fodder. *Opuntia* is native to Mexico and has spread through cultivation to many regions around the world, including South Africa even if has not yet been fully exploited in the livestock sector [6], Ethiopia, Australia, Italy, Spain, Argentina, Chile, and the Mediterranean basin [7]. *Opuntia* is a cactus plant fiber characterized both by insoluble and soluble fibers. Thanks to the soluble dietary fiber (pectin) that contributes to its physical properties and its high content in antioxidants (flavonoids, ascorbate), pigments (carotenoids, betalains), and phenolic acids it is appreciated also for its beneficial properties [8]. This review provides a comprehensive evaluation of *Opuntia* as an alternative feed that may potentially replace the traditional ones.

## 2. Botanical Characterization and Cultivation

The cactus *Opuntia ficus-indica* (L.) Mill., commonly called prickly pear or cactus pear, is a tropical and subtropical plant belonging to the dicotyledonous angiosperm *Cactaceae* family, which comprises about 1500 species of cactus [9]. It represents the most cultivated and widely known cactus species of the genus *Opuntia* that comprises about 300 species known in the world, of which only about 15 species are largely cultivated in the arid and semiarid regions for their edible fruits, their young cladodes as vegetables, and their cladodes as forage (Figure 2; [10]). Its cultivation requires less investment, and its yield can be higher than other crops grown in these areas, including cereals [10]. As this plant grows throughout the year, it generates feedstock almost continuously, thus eliminating the costs associated with silage preparation and storage. The production potential of cactus pear is at least 60 times higher than the productivity of pastures and is sufficient to produce forage to sustain five adult cows per year (more than 20 t DM/ha/year and provides 180 t/year of good quality water for livestock [10]). The fodder yield of *O. ficus-indica* has been compared to that produced by Oldman saltbush (*Atriplex nummularia* Lindl.) and Alfalfa (*Medicago sativa* L.) per unit of water received, showing higher values (3001.00 kg/ha vs. 944.80 kg/ha and 367.40 kg/ha, respectively [11]). The aboveground biomass (cladode and fruit) productivity of *O. ficus-indica* was greater than *Agave americana* L. with slightly lower water inputs (10.3 Mg DM/ha/year with a mean water input of 407 mm/year vs. 9.3 Mg DM/ha/year at 530 mm/year irrigation), demonstrating the most efficient use of water [12]. This plant can grow in arid and semiarid climates with a geographical distribution that includes Mexico, Latin America, South Africa, and Mediterranean countries. Italy represents the world’s third largest producer after Mexico and the USA [13]. 

The favorable conditions for its growth are a temperature ranging between 23 and 26 °C and full sunshine, although it can tolerate higher (up to 45 °C) and lower (down to −10 °C) temperatures for a few days. *O. ficus-indica* is considered a valuable source of nutrients in many of the world’s environments and is considered as valuable feedstuff in regions where other plants cannot survive due to extreme environmental conditions [14]. Due to the shape of its organs, which are like grass pads that store water in their tissue during the rainy season, this plant is able to adapt to extreme conditions. Another feature is represented by its shallow and extensive root system which allows the plant to take advantage of the scarce rainfall in arid environments. In places where water is a limiting factor for animal production, the high-water content of the cactus acts as an alternative to the water needs of animals. [15]. *Opuntia*, which is harvested from both wild and cultivated plantations to feed animals, is well consumed by them since it has been observed that all types of livestock, including cattle, sheep, goats, camels, and horses, prefer it to straw [16]. The major species of cacti used for livestock include *O. ficus-indica* Mill., *O. stricta* (Haw.) Haw., *O. lindheimeri* Engelm, *O. ellisiana*, *O. engelmannii* Salm Dyck, *O. chrysacantha* Berg, *O. amyclae*, *O. rastrera* Weber, and *Nopalea cochenillifera* Salm Dyck. The nutritional value varies according to the cactus species as recently reviewed [17]; in general, *Nopalea cochenillifera* Salm-Dyck. presents greater dry matter and soluble carbohydrate concentrations compared to cultivars of *O. ficus-indica*. 

From the agronomic point of view, *O. stricta* has presented less demand for nutrients, and it has been more tolerant to drought conditions than *N. cochenillifera* [18]. Nevertheless, *N. cochenillifera* has typically shown greater nutritive value than *O. stricta* (Haw.) Haw [19]. *Opuntia* species, however, are the most cultivated in the world for fodder production.

## 3. Chemical Composition

Cactus pear represents an excellent energy source, as it contains 61.7% of non-fibrous carbohydrates. The main constituent of *O. ficus-indica* cladodes is water (80–95%), followed by carbohydrates (3–7%), fiber (1–2%), and protein (0.5–1%) [20]. Several studies have reported its use as feed to supplement livestock diets due to its efficiency in converting water to DM, thus providing digestible energy [21,22]. 

The factors that mainly affect the chemical composition of cactus cladodes are the season, environmental factors, plant age, cladode order (position), cultivar, fertilization and harvest management, and planting density [17] (Table 1). Its nutritional value is similar to lettuce and spinach, while its dietary fiber is similar to other fruits and vegetables, such as mango, melon, apricot, grapes, spinach, artichoke, broccoli, and radish [14]. The site of cultivation and the physiological state of the cladode tissue may influence the mineral content of *O. ficus-indica* (Table 2).

Differences in the reported crude protein content could be related to growing site characteristics (slopes, poor soils), climatic conditions, and level of fertilization; therefore, the CP content of cactus is affected by factors such as soil moisture, soil N contents, and application of N fertilization. Variations in fiber values between studies can be due to differences in cactus species and cultivars and the stage of maturity. In particular lower NDF values may be related with the young age of cladodes used. Values of NDF and ADF may also vary according to the method of analysis used due to the high ash, pectin, and nonstructural carbohydrate contents of cactus. Moreover, factors such as soil mineral and moisture contents can significantly affect ash and mineral concentrations of spineless cactus. Water deficiency and high levels of Ca compounds in the soil tend to increase Ca in their cladodes [23]; whereas, salinity can affect the Na content [24]. 

The levels of phosphorus composition could perhaps be linked to varying degrees of soil fertility, cladode age, and climatic conditions [25], as cited by Tegegne [26].

**Table 1 animals-12-01597-t001:** Chemical composition (% DM) of *O. ficus-indica* according to different locations.

Location	DM	CP	NDF	ADF	CF	CHO	Reference
Portugal	13.75	7.3	18.51	10.70	-	63.7	[27]
Mexico	6.3	10.4	31.9	21.7	-	-	[28]
Italy	7.6	7.63	45.00	10.83	-	40.13	[29]
Ethiopia	9.74	2.73	20.7	2.30	11.6	60.9	[30]
Tunisia	13.5	3.8	3.51	-	8.5	58.2	[4]
Tunisia	19.1	2.97	27	15.5	-	-	[31]
Egypt	13	10.69	-	-	-	65.53	[32]

DM = dry matter; CP = crude protein; NDF = neutral detergent fiber; ADF = acid detergent fiber; CF = crude fiber; CHO = carbohydrate.

**Table 2 animals-12-01597-t002:** Mineral content (mg/100 g) of *O. ficus-indica* according to different locations.

Location	Plant Portion	K	Ca	Na	P	Fe	Mn	Reference
Mexico	Cladodes	2403	627	63	0.09	8.6	13.8	[33]
Mexico	Cladodes	6330	3440	30	390	22	8	[34]
Mexico	Cladodes	5469	1966	83.12	-	3.43	3.54	[35]
Brazil	Cladodes	3320	18.4	-	1.7	-	-	[36]
Tunisia	Cladodes	2350	5640	400	150	0.138	1.75	[37]
Tunisia	Cladodes	2370	9200	3100	40	13	2.73	[31]
Italy	Cladodes	5786	4876	80.28	-	17.09	45.12	[38]
Italy	Cladodes	6737	4941	79.28	-	16.63	57.32	[38]
Italy	Cladodes	5746	5933	79.93	-	15.82	55.76	[38]
Nigeria	Cladodes	-	250	-	-	25.00	36.00	[39]
Spain	Cladodes	224	177	1.71	16.38	0.130	0.78	[40]
Spain	Fruits	1583	26	0.625	-	0.198	0.303	[41]
Morocco	Seeds	305	481	48.33	1418	2.76	5.18	[42]
Algeria	Fruit peel	98	16	1.1	-	-	-	[43]
	Fruit pulp	199	12	1.09	-	-	-	
	Fruit seed	79	21	0.54	-	-	-	

## 4. Bioactive Compounds

The beneficial properties of *Opuntia* spp. are related also to their chemical content, such as polyphenols (Table 3). The content of phenolic compounds in *Opuntia* spp. is affected by several variables, such as maturity stage, harvest season, environmental conditions, and species. In particular, the fruits present a high content of flavonols that contribute to its antioxidant capacity. All parts of this cactus species contain phenolic compounds exhibiting antioxidant and anti-inflammatory properties, such as aromadendrin, taxifolin or dihydroquercetin, isorhamnetin, vitexin, kaempferol, quercetin, betalains, betacyanins, rutin, and isorhamnetin, as well as derivatives including myricetin and orientin [44,45]. Although the antioxidant and beneficial physiological effects of flavonoids and polyphenols have been extensively studied, very limited studies describe such mechanisms in animals [10].

## 5. Dietary Application of Opuntia in Animal Nutrition

During periods of drought, keeping animals in a hydrated state is difficult, especially in arid and semiarid areas. To address this problem, the high-water content in cactus pear cladodes could represent a solution [10]. In fact, due to the low DM content of cactus (11.69 ± 2.56%), diets formulated with large proportions of cactus roughage typically have a high degree of moisture, which may be beneficial in regions where water is scarce during certain seasons [59].

### 5.1. Effect on Animal Performance

Several studies have been performed to investigate the effect of *Opuntia* cladodes as forage on livestock performance and rumen physiology [18,60,61]. 

In a study conducted by Albuquerque et al. [62] in goats, the dietary inclusion of cactus pear silage up to 42% improved the eating and ruminating efficiency rates (the time spent eating and ruminating decreased from 248 to 160 min/day and from 414 to 303 min/day, respectively) and also improved body water retention, hence reducing drinking water intake without compromising animal performance. The basal diet of forty growing rams was replaced by cactus pear at 0, 20, 40, 60, and 80% on a dry matter basis [63]. The results of this study demonstrated that cactus pear could optimally substitute pasture hay to a level of 60% and additionally contributed to sustaining the water requirement of the animals. In the study conducted by Costa et al. [64] the voluntary intake of water decreased with increased levels of cactus pear in the diet of lambs, passing from 4.9 to 2.3 kg/day, demonstrating the importance of *O. ficus-indica* as a source of water in semiarid regions such as Brazil [64]. In other studies, it has been observed that when spineless cactus was supplied to sheep exceeded 300 g DM per day, animals consumed essentially no drinking water [15,65]. In Mexico, feeding *Opuntia* cladodes (untreated or protein enriched, from 500 g to 3 kg/day), was tested in ewes as a substitute for alfalfa hay (1.5 kg), during the last trimester of gestation to evaluate milk yield, birth weight, and lamb growth. Results showed that the enriched one did not show any improving effect, and that *Opuntia* can largely match the effects from a supplementation with alfalfa hay during the observed period showing similar birth weight (3.76 as average of CON and *Opuntia* groups); lambs fed both *Opuntia* experimental diets (untreated or protein-enriched) were on average heavier at weaning than the control group (alfalfa hay) (about 10 kg vs. 8 kg) representing a cost-saving option for industries based in arid and semiarid regions where forage supply is limiting [66]. The study conducted by Souza et al. [67] in Brazil evaluated the effect of three different addition ratios (0, 21, and 42%) of cactus silage on the carcass traits and meat quality of lambs. The experimental period was 84 days, with an adaptation period of 10 days, with a starting body weight equal to 19.8 kg. A higher weight at slaughter was observed for animals who received 42% cactus silage level (33.41 kg vs. 28.39 CON), probably connected with the higher diet intake found in this group. Lambs fed 42% cactus silage showed a larger rib eye area and a reduction in saturated fatty acids (C18:0 and C21:0), which is a positive characteristic from a health point of view. As mentioned above, the indiscriminate supplementation of cactus as fodder in ruminant feed could cause several critical issues, including diarrhea, decreased milk fat content, reduced DM consumption, and weight loss [60,68,69]. To address these problems, the combination of *O. ficus-indica* with other forages in the diets of ruminants has been evaluated in several studies with successful results. In dairy cattle farming, lactating Holstein cows fed a diet with different replacement levels (0, 12, 24, and 36%) of forage cactus (*O. ficus-indica* Mill.) for sorghum silage showed a quadratic effect of levels of forage cactus on the milk fat concentration, with a maximum milk fat of 4.08% with 24% of forage cactus [70]. In lambs, replacement of 33% maniçoba hay by spineless cactus can be recommended as optimal level, because it improved the fattening of the carcass, without causing negative effects on performance or meat quality [71]. Cardoso et al. [72] in a trial on lambs fed with increasing levels (0, 150, 300, and 450 g/kg of DM basis) of fresh spineless cactus found that the inclusion of the spineless cactus in the diet up to 450 g/kg of DM improved the microbial efficiency, nutrient utilization, and the growth performance of the lambs.

A significant reduction in water consumption (−16%) was found in lambs for whom saltbush and spineless cactus cladodes (1.7 to 1) replaced 60% of barley straw and 16% of the concentrate mixture (on a DM basis) compared to the control group [73]. In experimental diets performed on goats, the association of 74.9, 57.6, 37.0, and 7.2% of fresh spineless cactus with 8.4, 18.8, 31.2, and 48.3% of saltbush hay (more concentrate) was, respectively, tested. It was observed that the combination of 37% of fresh spineless cactus, 31.2% of saltbush hay, and 31.8% concentrate provided the highest final weight, daily gain, and total weight gain [74].

In arid and semiarid regions of the northeast of Brazil, where goat husbandry represents one of the important economic activities, it has been observed that the addition of 8.4% of saltbush (*Atriplex nummularia* L.) hay and 74.9% of *O. ficus-indica* can represent strategic alternative sources of fodder able to supply the nutritional deficiencies of these small ruminants [68]. The combination of these two sources of forage to the diet promoted a linear and quadratic effect (*p* = 0.02) on dry matter intake (DMI) and other nutrients. The diet composed of these percentages of saltbush and *O. ficus-indica* presented the lowest water balance (1.54 kg/day), water intake per kg of DM consumed (1.60 kg/day), and DM digestibility [74].

The study conducted by Ragab [75] reported that the inclusion of prickly pear peel (PPP) in the diets of male chicks (15 and 30%) improved their performance, since inclusion of PPP caused a significant (*p* < 0.05) decrease in live body weight gain (1061.1 g vs. 996.5 g) and a significant increase in feed intake (2736.2 g vs. 2885.2 g) during the total period of supplementation (from 14 to 70 days of age). Moreover, its integration, by minimizing the amount of expensive yellow corn grain needed in poultry diets, reduced the cost of feeding. 

The supplementation of 1% of *O. ficus-indica* as a source of dietary fiber in gilts during late gestation and lactation determined a variation in the biochemical parameters (increase in insulin and high density lipoproteins and a decrease in triglyceride concentration at gestation, farrowing, and lactation) and voluntary feed intake, which was higher compared to the control group [76]. Table 4 summarizes the application of *Opuntia* in animal nutrition and the main effects.

### 5.2. Effect on Products of Animal Origin

In the study conducted by Ortiz-Rodriguez et al. [77], the dietary supplementation of 14.9% of fresh *Opuntia-ficus indica* in Holstein cows improved both the quantity and quality of raw milk and fresh cheese. In fact, milk yield at 225 day of lactation was 10.864 L/day in the group supplemented with cactus pear, while in the control group this value was equal to 7.200 L/day. Additionally, the storage life of the fresh cheese also was improved, since the bacterial colony forming units (CFU) were significantly lower (*p* < 0.05) in the fresh cheese obtained from the group of cattle receiving the diet with *Opuntia* (4.0 CFU/mL vs. 5.1 CFU/mL; [77]). However, it cannot be ignored that both contents found were above that (2.0 CFU/g^−1^ [log10]) allowed by the Official Norm of the country in which the cheese was produced, which is indicative of poor hygiene and, therefore, not suitable for human consumption. The use of prickly pear byproducts (juice, peel, pulp, and seeds) as dietary supplement could be a way to enhance a more sustainable animal production system by reducing waste production. However, it was also reported that the poor shelf life of these products was due to their high levels of moisture and fermentable carbohydrates [88]. In fact, prickly pear byproduct outdoor storage is only possible for a few days due to bacterial fermentation [22]. It is advisable to ensile prickly pear byproducts with dry fodder or mature crop residues, such as wheat straw, to partially absorb the water, due to their high level of humidity. It has been observed that the silage obtained with 5% of straw is the best preserved, since it helps in achieving a significantly lower pH and ammonia nitrogen concentration. Moreover, the inclusion of only small amounts of straw is economically advantageous, as the cost on forages is greatly reduced [88].

The incorporation of *Opuntia* in ruminant diets also had effects on the quality of meat. The study conducted in Morocco by Otmani et al. [89] investigated the effect of the incorporation of cactus cladodes (CC; 30% concentrate diet) in the diet of Beni Arouss goat kids, during three months of a feeding trial starting from an average weight of 10.5 kg, showing higher proteins and lower fat and ash content in the CC group than the control (conventional supplementation). With reference to fatty acid, a decrease in C16:1 and an increase in C20:3n3 was observed in the *L. dorsi* of the CC group.

Pascoal et al. [90] conducted a study in which different levels (0%, 10%, 20%, and 30%) of forage cactus meal were added to the diet of rabbits in the growing phase to evaluate the effects of its inclusion on the productive performance, carcass characteristics, and economic evaluation. There was no effect of the dietary inclusion on average daily feed consumption, average daily weight gain, feed conversion, and the final weight of rabbits, suggesting that the nutritional quality of the diets were maintained as the level of inclusion of forage cactus meal increased. Equally, no effect was found on carcass weight, carcass yield, viscera weight, and liver weight.

The effect of substitution of yellow corn grains by prickly cactus pear (*O. ficus-indica*) fruit peel at the rates of 0, 5, 10, and 15% in broiler diets on the feed conversion ratio, meat quality, and its biological value was investigated by Badr et al. [91]. The results showed that the dietary inclusion of prickly cactus pear fruit peel enhanced the live body weight, feed intake, and feed conversion ratio (*p* < 0.05). The observed increments in live body weight and body weight gain (1765 g in the control group vs. 1865 g in 15% *Opuntia* integration) may be attributed to the increasing digestion of all nutrients, suggesting that the inclusion of this product in chick diets did not have a negative effect on diet palatability. Concerning the slaughter parameters, the dietary inclusion did not show any detrimental effect and achieved high carcass weight and dressing percentage, with a higher crude protein content (75.08% in the control group vs. 79.9% in the 15% *Opuntia* integration). Moreover, broilers fed diets containing 15% prickly pear products reached higher scores of taste, color, odor (aroma), texture, and overall acceptability.

## 6. Processing, Storage, and Quality Improvement of Opuntia and Byproducts for Animal Feeds

Fresh cladodes have always been used in livestock feed either whole or cut into strips or cubes. The cladodes are generally shredded or cut and partially dried before these are fed to animals. Regardless of the form in which it is used, the high-water content limits the handling and processing of fresh cactus pear cladodes as well as its intake. When dried, the addition of supplements corrects nutritional deficiencies in the composition of cladodes.

In order to increase the surface area for drying in the sun, fresh cladodes can be shredded by multi-purpose shredders into thin strips. In this way, the strips are dried in direct sunlight on drying floors and on racks slightly raised from the ground to ensure free air circulation.

The drying process can last 4–14 days depending on the thickness of the strips and the weather conditions [92]. Thereafter, they can be used in diet formulations. 

Cactus drying is a process mainly practiced in Brazil and South Africa. Under certain circumstances such as a short harvest period, the need to produce a commercial feed, or mix the cladodes with other ingredients, cactus cladodes are ground to produce a cactus meal. Care needs to be taken so as not to produce a very-fine meal that would increase transit time in the gut of ruminant animals [93]. Sun drying is a practical and cheap solution that avoids using expensive fuels that would increase production costs.

Only recently, prickly pear byproduct that comes from the fruits processed for juice extraction, comprising peel, pulp, and seeds has become available in large amounts, and it could be an interesting source for ruminant nutrition. 

Due to its high sugar content, bacterial fermentation quickly starts, and the use of a fermentative stabilizer is required. 

Concerning the conservation method, the addition of 100 and 150 g/kg of potassium metabisulfite, as preservative agent slightly slows down the early phase of the acidification process and limits the content of coliforms and *Enterobacteriaceae* after a three-week storage period [22]. 

A storage technique could be ensiling the mass with straw, and the best preservation seems to be obtained with 5% straw [88]. To make high-quality silage, appropriate levels of moisture (30–40%) and sugar in a full anaerobic environment are required in order to produce lactic acid fermentation [94]. Cactus cladodes contain sufficient carbohydrates for good lactic acid fermentation, but the high moisture requires careful mixing with chopped straw or bran [93].

A mixture of chopped cladodes (350 kg), olive cake (400 kg), and wheat bran (250 kg) with a pH of 4.5 could be used to replace oat hay partially or completely without affecting lamb performance and meat quality.

Moreover, cut-and-carry remains the most commonly used technique for cactus feeing. It prevents wastage and excessive grazing. Both spiny and spineless cactus cladodes can be chopped and mixed with other feeds, after harvesting and transporting to the barn.

Before chopping and feeding, the spines must be burnt. Different types of choppers are present in the market. In North Africa, cactus chopping is performed manually with knives and, in Ethiopia, transportable, low-cost choppers are made locally, and they are operated using human force. More sophisticated and motorized choppers are utilized in Brazil and Mexico. Two forms of cactus processing have been tested: the first has a knife with a larger dimension, approximately 12 cm × 5 cm, or the second processed in a fodder machine, with cuts of about 2 cm to expose the mucilage [95], maximize the DM intake, and avoid alterations in milk composition.

## 7. Conclusions

*Opuntia* cacti are efficient crops for arid regions since they can generate biomass under water stress conditions. Due to its water conservation, the prickly pear has been considered as a water source for animals. This provides an added value to conventional feeds. In addition, cladodes’ value due to their contents of minerals, proteins, dietary fiber, and phytochemicals (antioxidants) is a high nutritional quality. Since the prickly pear is a perennial plant, it can be used as a green resource mainly where herbage production is inadequate. It is also perceived that the partial replacement of imported feeds and fodder with *Opuntia* may help reduce farm costs and, therefore, enhance sustainability in such arid areas. Its inclusion in an animal’s diet can, therefore, have a positive impact on the reduction in water consumption and the costly importation of feeds, with a lower carbon footprint and a positive impact on the environment.

## Figures and Tables

**Figure 1 animals-12-01597-f001:**
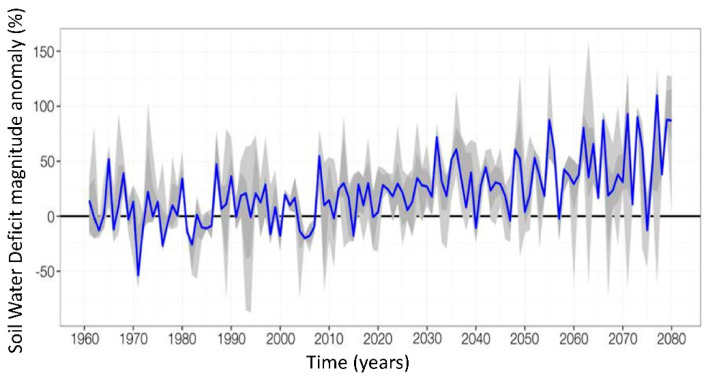
Water demand of the agricultural sector from 1980 to 2080 in relation to the soil water content deficit [2]. Reproduced with permission from Buffa and Ricciardi, 2017 [2].

**Figure 2 animals-12-01597-f002:**
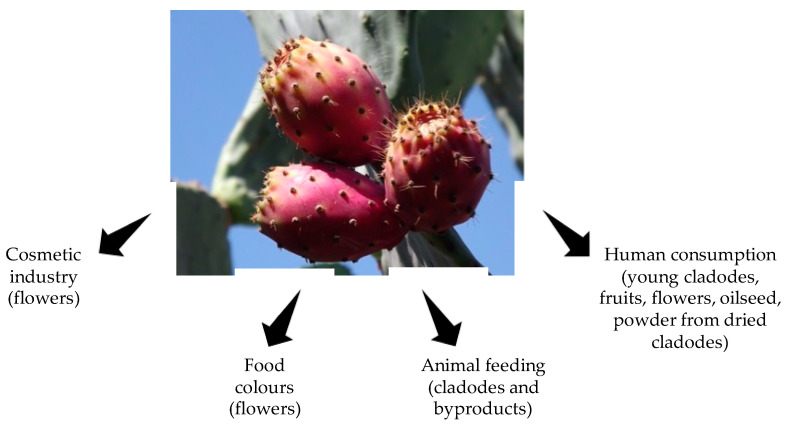
*O. ficus-indica* and its main uses.

**Table 3 animals-12-01597-t003:** Phenolic compounds (mg/100 g) present in *O. ficus-indica* portions according to different locations.

Location	Portion of the Plant	Type of Compound	Concentration (mg/100 g)	Reference
Egypt	Cladodes	TP	119.66 (GAE)	[32]
Egypt	Fruits	TP	123.56 (GAE)	[32]
Italy	Mature cladodes	TF	251 (DW; RU)	[46]
Italy	Immature cladodes	TF	678 (RU)	[46]
Italy	Purple fruits	TP	89.2 (FW; GAE)	[47]
Italy	Orange fruits	TP	69.8 (FW; GAE)	[47]
Italy	Fruit juice	FG	652.5 µg/mL	[48]
Spain	Cladodes	TPA	128.8 (FW)	[40]
Spain	Fruit pulp	TP	218.8 (FW; GAE)	[49]
		BE	40.6 (FW)	
		QU	9 (FW)	
		ISH	4.94 (FW)	
		LU	0.84 (FW)	
Spain	Fruit pulp	TP	2621.6 (GAE)	[50]
	Fruit peels	TP	4098.0 (GAE)	
	Fruit pulp	FL	62.87 (QE)	
	Fruit peels	FL	3228.0 (QE)	
Spain	Cladodes	ISH	258	[51]
		QU	42	
		KA	36	
		FA	43	
Mexico	Cladodes	TP	2690 (GAE)	[52]
Mexico	Fruit pulp	TP	285.3 (GAE)	[50]
Mexico	Fruit peels	TP	383 (GAE)	
Mexico	Fruit pulp	FL	74.99 (QE)	
Mexico	Fruit peels	FL	330.48 (QE)	
Morocco	Seed oil	FA	0.113	[53]
		4OHBA	0.198	
		FEA	0.446	
		VA	1.211	
		SYA	0.583	
Algeria	Cladodes	TP	2670 (GAE)	[54]
		FL	1186 (CE)	
Tunisia	Fruit pulp	TP	54.33 (GAE)	[55]
Tunisia	Flowers	GA	1.42	[45]
		PCA	11.83	
		TFA	16.49	
		RU	390.27	
		Q3OR	134.12	
		HYP	325.31	
Brazil	Fruit pulp	TP	12.34 (GAE)	[56]
		BE	3	
Brazil	Cladodes	TP	124–285 (DM; GAE) (dry period) 199–541 (DM; GAE) (rainy period)	[57]
		FL	153–302 (DM; QE) (dry period)	
			90–343 (DM; QE) (rainy period)	
Brazil	Fruit pulp	TP	719 mg GAE/L	[58]
		BTC	439.5 mg BE/L	

BEL = betalains; BE = betanin equivalent; BTC = betacyanins; CE = catechine equivalent; FA = ferulic acid; FEA = ferulhaldehyde; FG = flavonol glycosides; FL = flavonoids; FW = fresh weight; GAE = gallic acid equivalent; HYP = hyperoside; ISH = isorhamnetin; KA = kaempferol; LU = luteolin; PCA = p-coumaric acid; RU = rutin; SYA = syringaldehyde; TF = total flavonols; TP = total phenols; TFA = trans ferulic acid; TPA = total phenolic acids; Q3OR = quercetin-3-O-rhamonoside; QE = quercetin equivalent; QU = quercetin; VA = vanillin; 4OHBA = 4-OH benzaldehyde.

**Table 4 animals-12-01597-t004:** Use of *O. ficus-indica* in animal species.

Animal Species	Location	Portion of the Plant and Dose Diet	Duration of Integration	Effect	Reference
Dairy cattle	Mexico	Cladodes	80 days	↑ MY; ↑ organoleptic characteristics of fresh cheese; ↓ CFU in raw milk and fresh cheese;	[77]
Sheep	Brazil	Cladodes; 25, 50, 75, 100% DM basis	45 days	↓ voluntary intake of water; ↑ apparent digestibility of DM, OM, CP, NDF, and TC; ↓ DWG; no influence on digestibility of nutrients	[64]
Sheep	Tunisia	Cladodes; 3 kg	60 days	↑ plasma levels of calcium; no influence in MY; no influence on ovarian activity	[78]
Sheep	Mexico	Cladodes, 17% (fresh and dehydrated cactus)	11 weeks	No influence on performance; ↓ back fat of carcass	[79]
Goat	Tunisia	Cladodes; 3 kg	90 days	↑ consumption of *Opuntia ficus-indica* than *O. amyclae;*	[80]
Goat	Tunisia	Cladodes; ad libitum	84 days	No influence on milk production; ↓ carcass fat; higher C18:2 and CLA in meat; ↑ PUFA proportion in meat	[81]
Goat	Brazil	0%, 7%, 14%, 21%, 28%	60 days	↓ water intake; ↑ DM intake; ↓ milk fat; no influence on milk production	[69]
Rabbit	Egypt	Cladodes; 10%, 20%, 30%	9 weeks	↑ LBW, TWG and performance index (30% level); ↓ feed intake, protein intake, and digestible energy intake; ↑ digestibility coefficient and nutritive values (DM, CP, CF, TDN, DE); ↑ globulin concentration; ↓ cholesterol, LDL, HDL and total lipids; ↓ abdominal fat; ↑ plasma IgG and IgM	[82]
Rabbit	Egypt	Fruits and peel; 25%, 50%	8 weeks	No influence on growth performance; ↑ liver, heart and edible giblets weights; ↓ abdominal fat; ↑ TAC, GSH-Px, SOD, CAT; ↓ plasma triglycerides, cholesterol and LDL	[83]
Rabbit	Egypt	Peels; 10%, 20%, 30%	9 weeks	↑ LBW; ↓ feed intake, protein intake, digestible energy intake; ↑ plasma total protein, albumin and globulin; ↑ activity of intestinal amylase, lipase, protease; ↑ plasma IgM and IgG	[84]
Pig	Brazil	Not specified, 5%, 10%	15 days	No influence on feed conversion; no influence on the weight of organs (stomach, small and large intestine, liver, spleen, lungs, heart, kidneys)	[85]
Pig	Mexico	Cladodes; 1%, 1.5%, 2%	50 days	↑ feed intake during lactation; ↓ body weight loss at weaning; ↑ insulin concentration from day 100 of gestation to day 7 of lactation; ↓ glucose concentration; ↓ cholesterol and triglyceride levels (2%)	[86]
Broiler	Algeria	Cladodes; 5%, 10%	42 days	↓ plasma glucose, uremia, cholesterol and triglycerides; ↓ abdominal fat in carcasses; no influence on carcass weight	[87]

CAT = catalase; CFU = colony-forming units; CLA = conjugated linoleic acid; CF = crude fiber; CP = crude protein; DM = dry matter; DWG = daily weight gain; GSH-Px = glutathione peroxidase; LBW = live body weight; LDL = low density lipoprotein; MY = milk yield; NDF = neutral detergent fiber; OM = organic matter; PUFA = polyunsaturated fatty acids; SOD = superoxide dismutase; TAC = Total antioxidant capacity; TC = total carbohydrates; TDN = total digestible nutrients; TWG = total weight gain; ↑ = increase; ↓ = decrease.

## Data Availability

All data referred to in the manuscript are already published.

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
