# Peer review of "Opuntia spp. as Alternative Fodder for Sustainable Livestock Production"

_animals, 2022, doi:10.3390/ani12131597_

Round 1
Reviewer 1 Report
The theme chosen by the authors is very relevant in view of the present and future challenges for animal production in regions with scarcity of water and, after reading the manuscript, one is convinced that the genus Opuntia is a very valid alternative for feeding livestock in these regions.
The purpose of this manuscript is obviously not that of a systematic review on the potential of this cactus as a food for livestock, but, in the current form, one gets the impression that the topic deserved a more comprehensive and scrutinized research in the published literature, as well as more arranged data.
Line 59, it refers that the "review provides a quantitative evaluation" but this evaluation is limited to three tables, 1, 2 and 3.
The authors also mention in lines 61 and 62 that they intend to "review the nutritive value of Opuntia in food producing animals, examining performance and quality of derived products" but apparently fall short of these objectives.
Line 75, it says "produce forage to sustain five adult cows per year" but does not indicate the area, which I assume is one hectare.
Line 76, correct "180 t DM/ha/year of good quality water for livestock".
Line 110, correct "(Haw.) Haw [14]."
Line 113-114, I understand the meaning, but I suggest rewriting this sentence as it can cause confusion, especially after reading line 115, where it refers to "small amounts of carbohydrates (3-7%)".
Line 129, "4. Bioactive compounds", in this section refers only to polyphenols but does not indicate in which context of animal production or in which species they may be favorable.
Line 155, "5.1. Effect on animal performance", supposedly one of the most important sections of the manuscript boils down to two paragraphs, mixing chicks, small ruminants and gilts, almost randomly, without any organizational criteria.
Lines 211-213, it is referred "milk yield at 225 day of lactation was 10.864 l/day in the group supplemented with cactus pear, while in the control group this value was equal to 7.200 l/day". This important disparity deserved a more cautious scrutinize of the article from which this data was withdrawn. I read the full text of Ortiz-Rodriguez et al and it presents important weaknesses, such as, for example, the absence of data on water consumption in the two small groups during the dry season.
Lines 241-242, "The effect of substitution of prickly cactus pear (O. ficus-indica) fruit peel at rates of 0, 5, 10, and 15% with yellow corn grains in broiler diets", seems to me to be more the opposite by what is written below.
Line 252, I suggest replacing "degrees" with "scores".
Lines 326-328, rewrite the sentence.
Lin 331, reduction of water consumption instead of water resources?
Author Response
The theme chosen by the authors is very relevant in view of the present and future challenges for animal production in regions with scarcity of water and, after reading the manuscript, one is convinced that the genus Opuntia is a very valid alternative for feeding livestock in these regions.
The purpose of this manuscript is obviously not that of a systematic review on the potential of this cactus as a food for livestock, but, in the current form, one gets the impression that the topic deserved a more comprehensive and scrutinized research in the published literature, as well as more arranged data.
AU: We tried to include as much studies as possible so as to give a general picture on the possible use of Opuntia spp. based on several published studies.
Line 59, it refers that the "review provides a quantitative evaluation" but this evaluation is limited to three tables, 1, 2 and 3.
AU: We modified the sentence as follows: “This review provides a comprehensive evaluation of Opuntia as alternative feed that may potentially replace the traditional ones.” (Lines 67-68)
The authors also mention in lines 61 and 62 that they intend to "review the nutritive value of Opuntia in food producing animals, examining performance and quality of derived products" but apparently fall short of these objectives.
AU: The information provided was divided into three sections; one on the botanical characteristics and its cultivation, (2) the chemical composition of Opuntia (proximate analysis, mineral analysis and phenolics), (3) the effects on animal performance and (4) the effects of Opuntia on animal products. So perhaps, the statement may be rephrased to “In particular, the main objective is to review the nutritive value of Opuntia as a fodder crop, its effects on animal performance and its effects on animal-derived products.”
Line 75, it says "produce forage to sustain five adult cows per year" but does not indicate the area, which I assume is one hectare.
AU: It’s correct.
Line 76, correct "180 t DM/ha/year of good quality water for livestock".
AU: We corrected as follows: “180 t/year of good quality water for livestock”. (Line 82)
Line 110, correct "(Haw.) Haw [14]."
AU: The sour prickly pear is given this Latin name, which can be found in several botany books.
Line 113-114, I understand the meaning, but I suggest rewriting this sentence as it can cause confusion, especially after reading line 115, where it refers to "small amounts of carbohydrates (3-7%)".
AU: We modified as follows: “followed by carbohydrates (3-7%)” (Lines 121-122)
Line 129, "4. Bioactive compounds", in this section refers only to polyphenols but does not indicate in which context of animal production or in which species they may be favorable.
AU: We have added the following sentence: “Although the antioxidant and beneficial physiological effects of flavonoids and polyphenols have been extensively studied, very limited studies describe such mechanisms in animals [10].” (Lines 158-160)
Line 155, "5.1. Effect on animal performance", supposedly one of the most important sections of the manuscript boils down to two paragraphs, mixing chicks, small ruminants and gilts, almost randomly, without any organizational criteria.
AU: This section has been reorganized into four paragraphs: (1) reference to table 4 in which the effects of Opuntia on animal performance and rumen physiology are highlighted (2) a paragraph on small ruminants, (3) one on chicks and (4) one on gilts.
Lines 211-213, it is referred "milk yield at 225 day of lactation was 10.864 l/day in the group supplemented with cactus pear, while in the control group this value was equal to 7.200 l/day". This important disparity deserved a more cautious scrutinize of the article from which this data was withdrawn. I read the full text of Ortiz-Rodriguez et al and it presents important weaknesses, such as, for example, the absence of data on water consumption in the two small groups during the dry season.
AU: We agree with your reasoning, however, we considered that if the “G2 (n = 5) cows received the same diet as G1”, most probably the cows were given the same amount of water.
Lines 241-242, "The effect of substitution of prickly cactus pear (O. ficus-indica) fruit peel at rates of 0, 5, 10, and 15% with yellow corn grains in broiler diets", seems to me to be more the opposite by what is written below.
AU: You are right, the intention is the opposite, this has been rephrased to “The effect of substitution of yellow corn grains by prickly cactus pear (O. ficus-indica) fruit peel at rates of 0, 5, 10, and 15% in broiler diets” (Lines 288-289)
Line 252, I suggest replacing "degrees" with "scores".
AU: Done as suggested. (Line 299)
Lines 326-328, rewrite the sentence.
AU: We rewrite the sentence as follows: “Diets that contain prickly pear have a positive impact on the productive and reproductive systems in animal species, hence improving the sustainability of animal feeding. Since the prickly pear is a perennial plant, it can be used as a green herbage resource mainly where herbage production is inadequate.” (Lines 358-361)
Line 331, reduction of water consumption instead of water resources?
AU: We modified according to your suggestion. (Line 362)

Reviewer 2 Report
The revised paper has improved substantially.
After careful reading, I believe that only the issue of the English language should be looked at. Afterwards, I consider the paper ready to be published
Author Response
Manuscript animals-1721902
Response to Reviewer 2 Comments
After careful reading, I believe that only the issue of the English language should be looked at. Afterwards, I consider the paper ready to be published
AU: Efforts are being made to improve the orthography of the manuscript.

Reviewer 3 Report
The manuscript is interesting, well written and discussed. I would like to point out that the article is a continuation of other studies in this field on opuntia.
Currently, about 150 species have been identified, of which some of them were considered to be endangered. Fruits contain a lot of water over 92%, and the main component of Dry Matter are carbohydrates (approx. 50% DM). In addition, it is rich in water-soluble vitamins, known as phenolic, phytosterols, and lipids contain the most unsaturated fatty acids. The pro-health properties include moisturizing, stimulating the production of insulin by the pancreas, and the fiber-pectin contained in the fruit reduces cholesterol. Prickly pear is very popular in the Mexican diet.
The goal is quite clear and is to discuss the nutritional value and use of prickly pear in animal nutrition and its impact on the productivity and quality of products
Sugestion:
P:40-52 To specify which region it is about and which Agro production system.
I suggest using additional literature;
1. Opuntia spp. A strategic fodder and efficient tool to combat desertification in the WANA region. Ali Nefyaoui, Hichem Ben Salem.
2. Prickly Pear (Opuntia spp.) as an Invasive Species and a Potential Fodder Resource for Ruminant Animals. Nkosomzi Sipango, Khuliso Emmanuel Ravhuhali, Nthabiseng Amenda Sebola , Onke Hawu, Monnye Mabelebele, Hilda Kwena Mokoboki and Bethwell Moyo. Review: Sustainability 2022, 14, 3719. https://doi.org/10.3390/su14073719
3. The nutritional value and health benefits of fig prickly pear (Opuntia ficus-indica Mill.). Ewa Cieślik, Iwona Cieślik, Katarzyna Bartyzel. Post Fitoter 2016; 17(3): 213-217.
4. Opuntia spp.: Chemistry, Bioactivity and Industrial Applications. Mohamed Fawzy Ramadan, Tamer E. Moussa Ayoub, Sascha Rohn
5. Edible spineless cactus (Opuntia spp.): As alternative forage for animals in scarcity. Suresh Nipane et al., The Pharma Innovation Journal, SP 10(10), 421-424.
P:104-111 General phrases, what nutritional values, specify interspecies differences …
P:120-124 The discrepancy of the results of tables 1 and 2 should be presented in more detail
P:174-175 Report what the other groups were fed ?
P:179 Enter the type of fattening and the final weight of the lambs
P:185 Explain what parameters have been improved?
P:213-214 The CPU values ​​are influenced by factors related to storage conditions and hygiene, above all, and nothing was mentioned about them …
P:229 Complete in what fattening system (30%) and age for kids ?
P:324 Ideal crops???
P:334 Animal production traditional system
The discussion of the results and quotation of the literature complies with the requirements of the editorial office.
Summing up, I can say that due to the interesting results obtained, sometimes controversial and comparing many nutritional groups and parameters, the work should be accepted for publication after taking into account the reviewer's suggestions.
Author Response
Manuscript animals-1721902
Response to Reviewer 3 Comments
The manuscript is interesting, well written and discussed. I would like to point out that the article is a continuation of other studies in this field on opuntia.
Currently, about 150 species have been identified, of which some of them were considered to be endangered. Fruits contain a lot of water over 92%, and the main component of Dry Matter are carbohydrates (approx. 50% DM). In addition, it is rich in water-soluble vitamins, known as phenolic, phytosterols, and lipids contain the most unsaturated fatty acids. The pro-health properties include moisturizing, stimulating the production of insulin by the pancreas, and the fiber-pectin contained in the fruit reduces cholesterol. Prickly pear is very popular in the Mexican diet.
The goal is quite clear and is to discuss the nutritional value and use of prickly pear in animal nutrition and its impact on the productivity and quality of products
Suggestion:
P:40-52 To specify which region it is about and which Agro production system.
AU: If we are understanding well, the region is for arid and semi-arid areas and in the fodder production system.
We added as suggested, literature referred to regions in which Opuntia represents a potential fodder resource
I suggest using additional literature;
1. Opuntia spp. A strategic fodder and efficient tool to combat desertification in the WANA region. Ali Nefyaoui, Hichem Ben Salem (Added)
2. Prickly Pear (Opuntia spp.) as an Invasive Species and a Potential Fodder Resource for Ruminant Animals. Nkosomzi Sipango, Khuliso Emmanuel Ravhuhali, Nthabiseng Amenda Sebola , Onke Hawu, Monnye Mabelebele, Hilda Kwena Mokoboki and Bethwell Moyo. Review: Sustainability 2022, 14, 3719. https://doi.org/10.3390/su14073719 (Added)
3. The nutritional value and health benefits of fig prickly pear (Opuntia ficus-indica Mill.). Ewa Cieślik, Iwona Cieślik, Katarzyna Bartyzel. Post Fitoter 2016; 17(3): 213-217. (Not added in the paper since just abstract was in English)
4. Opuntia spp.: Chemistry, Bioactivity and Industrial Applications. Mohamed Fawzy Ramadan, Tamer E. Moussa Ayoub, Sascha Rohn (Added)
5. Edible spineless cactus (Opuntia spp.): As alternative forage for animals in scarcity. Suresh Nipane et al., The Pharma Innovation Journal, SP 10(10), 421-424. (Added)
P:104-111 General phrases, what nutritional values, specify interspecies differences
AU: We added a reference of review on this topic
P:120-124 The discrepancy of the results of tables 1 and 2 should be presented in more detail
AU: We added some consideration: Lines 132-144
P:174-175 Report what the other groups were fed?
AU: We corrected as follows: “lambs fed both Opuntia enriched or not were in average heavier at weaning than control (alfalfa hay) (about 10 kg vs 8 kg)” (Lines 196-198)
P:179 Enter the type of fattening and the final weight of the lambs.
AU: Done as suggested; we added the following sentence: “The experimental period was 84 days, with an adaptation period of 10 days with starting body weight equal to 19.8 kg.” (Lines 201-202)
P:185 Explain what parameters have been improved? Done
P:213-214 The CPU values are influenced by factors related to storage conditions and hygiene, above all, and nothing was mentioned about them …
AU: Right: we added some considerations (Lines 252-255)
P:229 Complete in what fattening system (30%) and age for kids?
AU: We added weight and length of the feeding trial. (Lines 269-270)
P:324 Ideal crops
AU: Substituted with “efficient” (Line 348)
P:334 Animal production traditional system
AU: added traditional
The discussion of the results and quotation of the literature complies with the requirements of the editorial office.
Summing up, I can say that due to the interesting results obtained, sometimes controversial and comparing many nutritional groups and parameters, the work should be accepted for publication after taking into account the reviewer's suggestions.

Reviewer 4 Report
This review evaluates the nutritive value of Opuntia genus and its use in livestock, as innovative alternative feed that may potentially replace the traditional ones.
The topic of paper is interesting and in the aim of the journal. However, paper requires major revision because needs some additional work and information to be considered for publication.
Specific comments and suggestions for improving the paper are:
I suggest changing the Keywords (Opuntia spp.; livestock; sustainability) that are already present in the title
Please, in the introduction insert something about opuntia (bioactive compounds, fiber, etc)
Line 64, please, after latin plant name insert the author citation
Line 77, 78, 80 as above
Line 122, please, check Opuntia species….in the table is Opuntia ficus indica
Table 3, please, insert in the concentration column unit measurement, standard used for TP, FL and TF.
Author Response
Manuscript animals-1721902
Response to Reviewer 4 Comments
This review evaluates the nutritive value of Opuntia genus and its use in livestock, as innovative alternative feed that may potentially replace the traditional ones.
The topic of paper is interesting and in the aim of the journal. However, paper requires major revision because needs some additional work and information to be considered for publication.
Specific comments and suggestions for improving the paper are:
I suggest changing the Keywords (Opuntia spp.; livestock; sustainability) that are already present in the title
AU: You are right. We added new keywords: prickly pear; arid regions; nutrition; animal performance (Line 27)
Please, in the introduction insert something about opuntia (bioactive compounds, fiber, etc)
AU: Added sentence on soluble dietary fiber and bioactive compounds: “Opuntia ficus-indica is a cactus plant fiber characterized both insoluble and soluble fibers. Thanks to the soluble dietary fiber (pectin) that contribute to its physical properties and the high content in antioxidants (flavonoids, ascorbate), pigments (carotenoids, betalains), and phenolic acids it is appreciated also for beneficial properties” (Lines 63-67)
Line 64, please, after latin plant name insert the author citation
AU: The cactus Opuntia ficus-indica (L.) Mill. (Line 70)
Line 77, 78, 80 as above
AU: Oldman saltbush (Atriplex nummularia Lindl.) and Alfalfa (Medicago sativa L.); Agave americana L. (Line 84 and 87)
Line 122, please, check Opuntia species….in the table is Opuntia ficus indica
AU: We modified the sentence as follows: “The site of cultivation, and the physiological state of the cladode tissue may influence the mineral content of O. ficus-indica (Table 2).” (Lines 129-131)
Table 3, please, insert in the concentration column unit measurement, standard used for TP, FL and TF.
AU: Inserted. In general, the units are mg/100g but there some instances where different units are specified

Round 2
Reviewer 1 Report
In terms of grammar, it seems to me that many sentences are missing commas, which compromises their meaning. For example, “In Mexico feeding Opuntia cladodes” (line 192), a comma is obviously needed between “Mexico” and “feeding”. Another example in lines 215-216: “In lambs replacement”.
Regarding the sentence in lines 55-56 that begins with “In India”, rewrite it, add more information and/or connect it with the surrounding text.
Line 107 “is easily consumed” after processing?
Line 134, (“slopes, poor soils” and?).
Line 198, rewrite “fed both Opuntia enriched or not were” to clarify the meaning.
Line 212, “In dairy farm”? Please specify.
Rewrite sentence in lines 327-329 to clarify its meaning.
Line 330, I suggest replacing “comes in” with “starts”.
Lines 361-362, although the authors refer a “positive impact on the productive and reproductive systems”, references in the text to the impact on the reproductive system of livestock species are very scarce or even contradictory (“no influence on ovarian activity”), as shown in Table 4 (reference #87).
Line 366, “could reflect” or “can have”?
Lines 366-367, "reduction", "reduction" and "reducing"...
In fact, I suggest rewriting the entire Conclusion as it is perhaps a bit simplistic.
Author Response
Manuscript animals-1721902 - Round 2: minor revisions
Response to Reviewer 1 Comments
In terms of grammar, it seems to me that many sentences are missing commas, which compromises their meaning. For example, “In Mexico feeding Opuntia cladodes” (line 192), a comma is obviously needed between “Mexico” and “feeding”. Another example in lines 215-216: “In lambs replacement”.
AU: We checked throughout the manuscript and added commas where appropriate.
Regarding the sentence in lines 55-56 that begins with “In India”, rewrite it, add more information and/or connect it with the surrounding text.
AU: We modified as follows:
“In India, the government has designated 30 percent of the country's surface as wasteland [5]. In fact, in hot dry zone of India, rural poor and smallholders are seriously affected by droughts and desertification. The future of the arid and semi-arid zones depends on the development of sustainable agricultural systems and on the cultivation of appropriate crops able to survive calamities, such as drought, extreme temperatures, and poor soils” (Lines 53-57).
Line 107 “is easily consumed” after processing?
AU: We changed “easily” in “well” (as reported by Souza et al., 2020) since it is a feed of excellent palatability (Line 108).
Line 134, (“slopes, poor soils” and?)
AU: We added details: “could be related to growing site characteristics (slopes, poor soils), climatic conditions and level of fertilization; therefore, CP content of cactus….” (Lines 133-134)
Line 198, rewrite “fed both Opuntia enriched or not were” to clarify the meaning.
AU: We modified the sentence as follows: “lambs fed both Opuntia experimental diets (untreated or protein-enriched) were ….” (Lines 197-198)
Line 212, “In dairy farm”? Please specify.
AU: We specified “In dairy cattle farming, lactating Holstein cows” (Line 213).
Rewrite sentence in lines 327-329 to clarify its meaning.
AU: We rewrote the sentence as follows: “Only recently, prickly pear by-product that comes from fruits processed for juice extraction, comprising of peel, pulp, and seeds is available in a large amount and it could be a very interesting source for ruminant nutrition” (Lines 326-328).
Line 330, I suggest replacing “comes in” with “starts”.
AU: Done as suggested (Line 329).
Lines 361-362, although the authors refer a “positive impact on the productive and reproductive systems”, references in the text to the impact on the reproductive system of livestock species are very scarce or even contradictory (“no influence on ovarian activity”), as shown in Table 4 (reference #87).
AU: We deleted this sentence and modified the whole conclusions paragraph according to the suggestion.
Line 366, “could reflect” or “can have”?
AU: We corrected in “can have”
Lines 366-367, "reduction", "reduction" and "reducing"...
AU: We rewrite the paragraph of conclusions.
In fact, I suggest rewriting the entire Conclusion as it is perhaps a bit simplistic.
AU: We rewrite the paragraph of conclusions, according to Reviewer’s suggestion.

Reviewer 4 Report
Dear authors, the answer regarding question on table 3 is incomplete!! Standard used for TP and FL are missing in:
Italy orange fruit TP
Spain fruit peels TP, fruit pulp FL, fruit peels FL (unit measurament is µg/g)
Algeria cladode TP, FL
Tunisia fruit pulp TP
Please add them!
Author Response
Manuscript animals-1721902 - Round 2: minor revisions
Response to Reviewer 4 Comments
Dear authors, the answer regarding question on table 3 is incomplete!! Standard used for TP and FL are missing in:
Italy orange fruit TP
Spain fruit peels TP, fruit pulp FL, fruit peels FL (unit measurament is μg/g)
Algeria cladode TP, FL
Tunisia fruit pulp TP
Please add them!
AU: We are sorry for the oversight; missing standards have been added in Table 3.

This manuscript is a resubmission of an earlier submission. The following is a list of the peer review reports and author responses from that submission.
Round 1
Reviewer 1 Report
General Comments
The review is of little benefit and contains a lot of references to questionable data. Typically, proponents of hydroponically grown fodder do not conduct studies with adequate Control groups for comparison. I agree that in situations where there is a shortage of available land to grow fodder, intensively grown fodder can be beneficial. The authors should re-write the manuscript showing much more caution about potential benefits of hydroponically grown fodder and a more realistic review of the actual costs. Some specific comments about emotive language and poor referencing are shown below.
.
Specific Comments
It is difficult to interpret Figure 1 without more context?
The writing style is too emotive. For example:
Lines 54-56
The hydroponic fodder system is an ideal solution in places where there is limited land area for the growing of fodder or where pasture grazing is limited or non-existent. It is also sustainable as it occupies a small land area, thus making it ideal for limited areas.
Similar in Lines 85-86
The optimum conditions for the germination of seeds and their development into high quality and highly nutritious animal feeds, include a hygienic environment free of chemicals like insecticides, herbicides, fungicides and artificial growth promoters [11].
Also, Line 163
Hydroponic fodder is highly succulent.
.
Why are hygienic and chemical free environments optimum for producing ‘highly nutritious’ feed?
Table 1 needs a comparison to other feed sources, such as grain without sprouting.
A number of articles referred to in terms of production benefits are questionable at best. Rahim 2015 did not have a proper control group (whereby the authors should have fed the barley grain prior to sprouting in the TMR Control group). Furthermore, Abd Rahim did not assess the dry matter intake of green fodder by feedlot cattle, as indicated in lines 184-185.
‘the dry matter intake 184 of green fodder by feedlot cattle and dairy cattle were low due to its high moisture content’
Author Response
Manuscript animals-1557751
Response to Reviewer 1 Comments
The review is of little benefit and contains a lot of references to questionable data. Typically, proponents of hydroponically grown fodder do not conduct studies with adequate Control groups for comparison. I agree that in situations where there is a shortage of available land to grow fodder, intensively grown fodder can be beneficial. The authors should re-write the manuscript showing much more caution about potential benefits of hydroponically grown fodder and a more realistic review of the actual costs. Some specific comments about emotive language and poor referencing are shown below.
AU: Thanks for your comments and for giving us the opportunity to improve our work. Additional considerations concerning costs of hydroponic fodders were added in the text (Lines 144-161).
Specific Comments
It is difficult to interpret Figure 1 without more context?
AU: Agreed. Additional information was added within the text. (Lines 41-42)
The writing style is too emotive. For example:
Lines 54-56
The hydroponic fodder system is an ideal solution in places where there is limited land area for the growing of fodder or where pasture grazing is limited or non-existent. It is also sustainable as it occupies a small land area, thus making it ideal for limited areas.
AU: Agreed. The word “ideal” was replaced in both instances to provide a more pragmatic approach.
The sentence has been modified as follows: “The hydroponic fodder system is a potential solution in places where there is limited land area for the growing of fodder or where pasture grazing is limited or non-existent. It is also sustainable as it occupies a small land area, thus making it feasible for limited areas.” (Lines 56-58)
Similar in Lines 85-86
The optimum conditions for the germination of seeds and their development into high quality and highly nutritious animal feeds, include a hygienic environment free of chemicals like insecticides, herbicides, fungicides and artificial growth promoters [11].
AU: We are not trying to correlate the ‘highly nutritious animal feeds’ to a ‘hygienic environment free from….’
“The optimum conditions, for the germination of seeds and their development into high quality and highly nutritious animal feeds, include a hygienic environment free from chemicals like insecticides, herbicides, fungicides and artificial growth promoters”. (Lines 87-89).
Also, Line 163
Hydroponic fodder is highly succulent.
AU: Agreed. This has been modified to read as follows: “As opposed to hay, straw and concentrates, the fresh hydroponic fodder is more succulent…” (Line 202).
(The high-water content is linked to its freshness as compared to feeds offered to most housed animals, that do no pasture)
Why are hygienic and chemical free environments optimum for producing ‘highly nutritious’ feed?
AU: A comma is inserted after the word “conditions” to improve the flow of the sentence as shown in the amendment above. (Lines 87-89). “The optimum conditions, for the germination of seeds and their development into high quality and highly nutritious animal feeds, include a hygienic environment free from chemicals like insecticides, herbicides, fungicides and artificial growth promoters”.
Table 1 needs a comparison to other feed sources, such as grain without sprouting.
AU: The aim of Table 1 was to compare the proximate analysis for hydroponic barley reported in a number of studies.
A number of articles referred to in terms of production benefits are questionable at best. Rahim 2015 did not have a proper control group (whereby the authors should have fed the barley grain prior to sprouting in the TMR Control group). Furthermore, Abd Rahim did not assess the dry matter intake of green fodder by feedlot cattle, as indicated in lines 184-185.
AU: It is true that Rahim 2015 did not compare barley grain to sprouted barley in the TMR. However, evident differences would have been noted as Ortiz et al., 2021 reported that the moisture contents were 88.5 and 788 g/kg respectively, while the sprout biomass on dry basis were 32.5 and 935.8 g/kg respectively. Most probably Rahim and co-authors followed the conventional feed regimen and compared this to a regimen with added sprouted barley.
Ortiz, L. T., Velasco, S., Treviño, J., Jiménez, B., & Rebolé, A. (2021). Changes in the Nutrient Composition of Barley Grain (Hordeum vulgare L.) and of Morphological Fractions of Sprouts. Scientifica, 2021.
‘the dry matter intake 184 of green fodder by feedlot cattle and dairy cattle were low due to its high moisture content’
AU: Agreed. Considering that wheat bran (WB) and straw have a lower moisture content than sprouted barley (BS), a composite percentage of WB/BS of 55% is much more superior in dry weight than 15% WB and 40% HB.
The following was added after the sentence: ‘when compared to the control ration with a high dry matter content’ –showing this relationship. (Line 227)

Reviewer 2 Report
This review article simultaneously addresses two topics that are increasingly relevant today, i.e., alternatives for producing food for farm animals in smaller areas of land and with less water consumption. However, the way the authors approach the topic is, in my view, quite unusual and debatable as it combines a forage production method, hydroponics, with an alternative forage species, Opuntia spp., two practically independent themes with enough published scientific literature to substantiate a review article by themselves. One could also question the non-inclusion in this review of other forage candidates for regions with less water resources, such as Klein grass and Rhodes grass.
In addition to the productive performance of several livestock species that ingest forage produced by hydroponics and Opuntia spp., which the authors focus on, I think it would be very important to know their biomass yields, also comparing them with other forage crops and production methods. Without this information, it is difficult to determine whether these are viable alternatives as, e.g., even a forage with high nutritional value may not be suitable to feed livestock if its biomass productivity is limited, which may force a drastic reduction of the livestock numbers in a certain area.
Therefore, I suggest choosing only one of the topics of the article, hydroponics or Opuntia spp., and delving into the chosen one, namely with information on the biomass yields, since only then will you have some idea about the feasibility of these alternatives for feeding livestock. If the option is Opuntia spp., I also suggest comparing this genus with other species for the production of forage in regions with less water resources.
Author Response
Manuscript animals-1557751
Response to Reviewer 2 Comments
This review article simultaneously addresses two topics that are increasingly relevant today, i.e., alternatives for producing food for farm animals in smaller areas of land and with less water consumption. However, the way the authors approach the topic is, in my view, quite unusual and debatable as it combines a forage production method, hydroponics, with an alternative forage species, Opuntia spp., two practically independent themes with enough published scientific literature to substantiate a review article by themselves. One could also question the non-inclusion in this review of other forage candidates for regions with less water resources, such as Klein grass and Rhodes grass.
AU: We agree with the reviewer that not enough information has been provided to show the importance of both fresh forage types.
Perhaps, this point has to be clarified by providing more insight into the two types of fresh fodder. We carried out a multivariate meta-analysis which we did not present in our first submission. We intend to justify the mention of Opuntia alongside hydroponic fodder. See Lines 524-533 and Figure 3.
In addition to the productive performance of several livestock species that ingest forage produced by hydroponics and Opuntia spp., which the authors focus on, I think it would be very important to know their biomass yields, also comparing them with other forage crops and production methods. Without this information, it is difficult to determine whether these are viable alternatives as, e.g., even a forage with high nutritional value may not be suitable to feed livestock if its biomass productivity is limited, which may force a drastic reduction of the livestock numbers in a certain area.
AU: We tend to agree with the review on this.
We thoroughly searched for literature and we attempted to include as much information as possible. Additional information on biomass yield was added. See Lines 165-182 for hydroponics and Lines 378-391 for Opuntia productivity.
Studies that we quote mainly focus on animal production systems, considering minimally the cultivation methods and potential improvement of the fodder crop under hydroponic cultivation.
Therefore, I suggest choosing only one of the topics of the article, hydroponics or Opuntia spp., and delving into the chosen one, namely with information on the biomass yields, since only then will you have some idea about the feasibility of these alternatives for feeding livestock. If the option is Opuntia spp., I also suggest comparing this genus with other species for the production of forage in regions with less water resources.
AU: Although, we understand the reviewer’s comment to segregate the two from each other, we carried out the above-mentioned multivariate meta-analysis that may bring these two culture types together; i.e. the fresh green hydroponic fodder and the succulent Opuntia.

Reviewer 3 Report
We do not recommend this paper due to some aspects.
Data on opuntia and others genotypes in animal feed is very well clarified in several recently published reviews.
I agree that hydroponic crops produce high quality fooder.
The review is very well prepared, but it only shows the nutritional value of hydroponic foods, but nothing that leads us to conclude that they could be as alternative fodder for sustainable livestock production.
hydroponic fodder requires a lot of labor, water, a lot of fertilization, that is, it is very costly. It was researched a lot, but it has no applicability.
The authors themselves comment on this: "Although some studies were conducted decades ago".
Author Response
Manuscript animals-1557751
Response to Reviewer 3 Comments
We do not recommend this paper due to some aspects.
Data on opuntia and others genotypes in animal feed is very well clarified in several recently published reviews.
I agree that hydroponic crops produce high quality fooder.
The review is very well prepared, but it only shows the nutritional value of hydroponic foods, but nothing that leads us to conclude that they could be as alternative fodder for sustainable livestock production. Hydroponic fodder requires a lot of labor, water, a lot of fertilization, that is, it is very costly. It was researched a lot, but it has no applicability.
The authors themselves comment on this: "Although some studies were conducted decades ago".
AU: We respect the opinion of this reviewer, but we tend to disagree on a number of points.
We did a thorough literature search of the information within the scientific domain, but the information is scattered and incomplete.
The intention of this review is to bring awareness on what is already present and what needs to be done. The analysis of fodder and Opuntia was intended to bring out their value as fresh additives to the conventional feed regimen/ration given to a number of farm animals. With proximate values at hand, it would be possible to work out the value of such fresh fodder within the TMR.
We disagree on the statement by the reviewer stating that ‘hydroponic fodder requires a lot of labor, water, a lot of fertilization, that is, it is very costly’. In our opinion, all cultivation practices may be labour intensive but growing fodder in trays within a small space has its own benefits (i.e. can be done on farm, reducing the carbon footprint of fodder/feed importation and importation costs!), water is generally recycled within the system (besides we are dealing with fresh material in semi-arid and arid regions), fertilization is more costly in grain cultivation (with hydroponics, the right fertilization reduces significantly costs, losses and over-use). The cost of hydroponic fodder has been considered by some studies. In one study, the costings of conventional feed ration were Eur 411 per ton as compared to hydroponic feed ration Eur 290 per ton [24].
Cost considerations were also added according to Reviewer 1.

Reviewer 4 Report
Please revise some minor errors indicated in the manuscript. Also the tables need need to be revised to make them easy to understand

Author Response
Manuscript animals-1557751
Response to Reviewer 4 Comments
Please revise some minor errors indicated in the manuscript. Also, the tables need to be revised to make them easy to understand.
AU: We thank the reviewer for the time spent on our manuscript. We checked throughout the manuscript and corrected the errors as suggested; we also improved the tables.

Round 2
Reviewer 1 Report
The manuscript does not accurately reflect the feeding value for hydroponically grown fodder compared with grain. Many of the results are misleading, as they are not presented on a Dry Matter basis or equivalent energy basis.
Reviewer 2 Report
The information added by the authors on the productivity and profitability of hydroponics was scarce and not very convincing in terms of applicability because e.g. no comparison of more reliable productivity parameters was carried out, to put conventional methods on an equal footing.
As for me, the combination of these two topics, a forage production method and Opuntia, still doesn't make much sense, lacking a more convincing contextualization that even the added figure 3 cannot substantiate.
Therefore, I continue to think that the authors should write a more in-depth review on just one of the topics or else cover in a more comprehensive review other species and production methods with potential utility in regions with water and land scarcity.
Reviewer 3 Report
The paper contributes absolutely nothing to the sustainability of production systems, as far as Hydroponic fodder is concerned.
Opuntia has already been much discussed